# Can Jurors Disregard Inadmissible Evidence? Using the Multiphase Optimization Strategy to Test Interventions Derived from Cognitive and Social Psychological Theories

**DOI:** 10.3390/bs15010007

**Published:** 2024-12-26

**Authors:** Pamela N. Sandberg, Tess M. S. Neal, Karey L. O’Hara

**Affiliations:** 1New College of Interdisciplinary Arts and Sciences, Arizona State University, Glendale, AZ 85306, USA; psandber@asu.edu (P.N.S.);; 2Department of Psychology, Iowa State University, 1347 Lagomarcino Hall, 901 Stange Rd., Ames, IA 50011, USA

**Keywords:** juror, inadmissible evidence, mental control, psychological reactance, prejudicial, multiphase optimization strategy

## Abstract

Inadmissible evidence generally biases jurors toward guilty verdicts; jurors who hear inadmissible evidence are more likely to convict than jurors not exposed to inadmissible evidence—even when *admissible* evidence is constant. When inadmissible evidence is introduced, the common legal remedy is judicial instructions to jurors to disregard it. Appeals courts repeatedly affirm instructions to disregard as a sufficient safeguard of defendants’ constitutional rights, despite research finding that jurors do not disregard when instructed. The goals of this research were to (1) test the main and interactive effects of four theory-driven candidate strategies to help jurors disregard inadmissible evidence (i.e., inducing suspicion, giving a substantive reason for disregarding, committing to disregarding, advising future jurors) and identify an optimized intervention package, and (2) evaluate whether adding the optimized intervention package showed more favorable effects than judicial instructions only. Study 1 used a 2^4^ full factorial randomized controlled trial to evaluate the four candidate intervention strategies. A synergistic interaction among the candidate components suggested an optimized intervention package comprising all four interventions. Study 2 used a parallel four-arm randomized controlled trial to compare conviction rates in the same hypothetical murder trial under four conditions: (1) no exposure to inadmissible evidence, (2) exposure to inadmissible evidence without objection, (3) exposure to inadmissible evidence + judicial instructions (“standard practice”), and (4) exposure + judicial instructions + optimized intervention package. Across both studies, mock jurors who received the optimized intervention package returned significantly lower conviction rates than comparison conditions. These findings show early promise that novel intervention strategies may assist jurors in disregarding inadmissible evidence. Interpretation, limitations, and calls to action are discussed.

## 1. Introduction

John Antonio Poole was eighteen years old when he was sentenced to life in prison without the possibility of parole. The sentence was mandatory under the sentencing scheme Michigan had in place at the time ([31]). Appealing the verdict, Poole asked the court to find that he had ineffective assistance of counsel at trial because his attorney failed to object to inadmissible evidence—a hearsay statement made by Poole’s uncle that implicated Poole while the uncle was attempting to negotiate the settlement of his own charges through a plea bargain. The appeals court determined that this was not ineffective assistance of counsel, “because if the trial court had instructed the jury to disregard the [inadmissible] remark, it likely would have drawn the jury’s attention to [it]” ([31]). After recognizing the futility of this most common legal remedy, the court denied Poole’s petition to appeal his sentence. Poole was convicted by a jury exposed to inadmissible evidence without so much as an objection from his own attorney. His attorney’s silence was then ratified by the appellate court reviewing his case.

Every day, criminal defendants across the United States face punishment—up to and including the death penalty—on the basis of information that enters a courtroom. There are legal rules for the admissibility of evidence, and judges are trained to respond to inadmissible evidence by instructing the jury to disregard it. Decades of research have consistently found that jurors are either unable or unwilling to follow judicial instructions to disregard inadmissible evidence (see, e.g., [39]; [40]), yet the legal system continues to rely heavily on jurors’ ability to disregard inadmissible evidence.

## 2. How Evidence Enters the Court

Evidence presented in court must adhere to specific legal rules and procedures with the goal of facilitating just outcomes, which may evolve over time. At the start of a criminal case, the prosecutor and defense attorney must disclose to one another the evidence they intend to introduce at trial, to allow time to respond to the evidence (see, e.g., [12]). These disclosures can also raise issues about the admissibility of the evidence that each side proffers. There are particular rules within the law about what types of evidence are admissible—and which are not—based on theories of procedural and substantive justice. Before trial, an attorney may ask the court to make evidentiary rulings about the admissibility of evidence the other side intends to offer at trial. The goal of pretrial rulings is to prevent jurors from being exposed to inadmissible evidence that could affect their decisions.

The primary source of guidance for judges on admissibility issues are the aptly titled rules of evidence. The Federal Rules of Evidence apply in federal courts. State courts have their own rules of evidence, but most are modeled on the federal rules. The overarching standard of admissibility is contained in ([13]), which states that the probative value of evidence must outweigh its prejudicial impact in order to be admissible. This balancing test is the prevailing method attorneys use to argue for or against admissibility, and it is also among the most common rules used by judges to make determinations of admissibility. In practical terms, this means that a judge has an obligation to exclude evidence that makes it likely the jury will convict the defendant for reasons other than the evidence presented at trial.

Despite safeguards offered by rules of evidence and pretrial rulings, inadmissible evidence does make its way into courtrooms. A witness might, for example, accidentally refer to inadmissible evidence such as prohibited hearsay statements. When this happens, the opposing attorney can (1) object to the testimony, (2) request it be stricken from the record, (3) ask the court to instruct the jury to disregard the inadmissible testimony, or (4) ask for a mistrial (see [32]).

In egregious cases of highly prejudicial evidence, attorneys can ask the court to declare a mistrial, though courts have been extremely hesitant to utilize this costly remedy. Appeals courts throughout the U.S. have upheld the presumed efficacy of instructions to disregard inadmissible evidence. In 2023 alone, such appeals were denied in Maryland ([41]), Texas ([44]; [33]), California ([28]), Wisconsin ([36]), Oregon ([38]), and Pennsylvania ([10]). This is not a comprehensive list.

When judges instruct jurors to disregard inadmissible evidence, they generally expect jurors to behave as if they never heard such evidence (that is, they expect and assume the admonition eliminates any potential increase in conviction rates that might otherwise occur after exposure to inadmissible evidence). However, extensive psychological studies reveal that these instructions often do not achieve the desired effect in reality (see, e.g., [35]; [40]; [42]).

## 3. The Problem of Inadmissible Evidence

Courts impose hefty sentences on defendants who have been convicted by juries exposed to inadmissible evidence, even de facto life sentences (see, e.g., [16]; affirming an eighty-year prison sentence issued by a jury instructed to disregard the prosecutor’s comment about the defendant’s invocation of his right to silence). Legal experts, courts, and researchers are well aware of this issue. Many scholars question the efficacy of jurors following judges’ instructions to ignore such evidence (e.g., [17]). Some courts suggest that a defendant’s constitutional rights might be better upheld by forgoing the typical remedy of a “disregard” instruction altogether. Recall from the introductory case that John Poole’s appeal was denied on the assumption that asking for an instruction to disregard would only draw further attention to the inadmissible evidence ([31]).

Even the Supreme Court has struggled to create a workable rule for addressing the problem of inadmissible evidence. In [11] ([11]), the Court affirmed a general rule that instructions to disregard are an appropriate “cure” when juries are exposed to inadmissible evidence. But the Cruz Court also recognized that the risk of jurors ignoring instructions to disregard was too great with evidence as inflammatory as a co-defendant’s confession, so this became the single exception. The Court did not, however, explain why the risk would be less with other types of evidence (e.g., evidence collected without a warrant, prior convictions outside the admissible timeframe, or prohibited hearsay statements about the defendant’s guilt). Nor did the Court attempt to create guidelines that would help protect future defendants from other types of inadmissible evidence.

Similar problems exist under state laws. In Illinois, for example, courts consider such factors as the character and magnitude of the inadmissible evidence on a case-by-case basis ([29]). This discretionary system allows judges to produce highly variant outcomes on the basis of different considerations. When a legal issue is not preserved for appeal at trial, Illinois judges must speculate about how much impact the inadmissible evidence had upon the jury. If the court agrees that the evidence is “closely balanced,” an appeal is still allowed to preclude any possible argument that an innocent person was wrongfully convicted ([26]). Therefore, the law requires the judge to (1) estimate the decisiveness of the trial verdict (i.e., determine if the admissible evidence alone was robust enough to likely secure a conviction without the inadmissible evidence) and (2) gauge the influence of the inadmissible evidence on the jurors’ conviction certainty based on the admissible evidence.

The problem is not limited to Illinois: in Arizona, appellate judges must determine “the probability that the jurors, under the circumstances of the particular case, were influenced by the remarks” ([37]). This standard again calls for a judge’s interpretation about whether inadmissible evidence had “too much” impact—without specifying what that amount might be. Other notable decisions were made by the California Supreme Court (holding that evidence of inadmissible prior child molestation was not prejudicial where the incidents were “relatively minor” and “largely cumulative” of many related incidents; [25]) and the Supreme Court of New York (holding that, though the jury heard inadmissible statements about the defendant’s addiction, they did not “clearly or inexorably paint the defendant as an addict who repeatedly stole money from her parents;” [27]).

## 4. Psychological Underpinnings of Conviction Bias Induced by Inadmissible Evidence

The issues raised by legal scholars and illustrated through case law are supported by psychological research and theory. For example, findings from the University of Chicago jury project in the 1950s found that instead of ignoring inadmissible evidence, mock jurors did “the exact opposite of what they were told” (described by [40]). The admonition given by the judge to disregard inadmissible evidence actually *increased* its prejudicial effect: mock jurors who heard an instruction to disregard evidence of the defendant’s insurance awarded *higher* verdicts than mock jurors who heard about the defendant’s insurance without objection. The researcher concluded that verdicts would be lower if “no such fuss” (an objection) was made ([3]). Many subsequent studies have confirmed that jurors are unable or unwilling to disregard inadmissible evidence ([40]), and a later meta-analysis found that criminal defendants are more likely to be convicted when jurors hear inadmissible evidence, despite objections from their lawyers ([35]).

Psychologists have spent decades working to better understand this phenomenon. Methods of improving juror comprehension of judicial instructions has been a popular research topic for decades (e.g., [22]; [23]; [34]). While this research addresses an important problem, much of it relies on the assumption that jurors simply do not understand a judge’s instruction to disregard inadmissible evidence. However, the psychological processes at work are more nuanced than simple misunderstanding. Other social, cognitive, and social–cognitive theories have also been tested, and two have emerged as the most popular explanations for jurors’ failure to disregard inadmissible evidence.

The theory of the Ironic Processes of Mental Control ([43]) holds that, under any conditions which reduce cognitive capacity, a person may experience enhanced “sensitivity to mental contents that are the ironic opposite of those that are intended” ([43]). This effect has been demonstrated in the context of juror instructions to disregard inadmissible evidence ([8]). The theory predicts that jurors’ cognitive capacities are reduced by the effort to recall and follow extensive instructions from a judge, while also following the complex evidence and arguments made by attorneys on both sides of a case. This strain on cognitive capacity enhances their sensitivity to the contents of evidence they are instructed to disregard. The theory has been supported by empirical evidence, in which higher cognitive loads were associated with more responsiveness to inadmissible evidence in a Stroop task ([8]).

The second prevailing explanation of why jurors find inadmissible evidence compelling is the theory of psychological reactance ([2]), which holds that people are motivated to regain behavioral freedoms when these freedoms are reduced or threatened. In the context of jury decision-making, the theory predicts that jurors will react to a judge’s instructions to disregard by being even more attentive to inadmissible evidence they are instructed to ignore. Jury research has shown that jurors do indeed react to instructions to disregard inadmissible evidence—especially when the admonition is strong ([19]).

## 5. Theory-Driven Strategies to Reduce Conviction Bias Induced by Inadmissible Evidence

A theoretical understanding of this problem is a critical first step, but equally important is the application of theory to practice. Researchers have proposed and tested many different interventions on mock jurors in laboratory settings. One group of researchers was able to successfully overcome ironic processes of mental control and psychological reactance by making jurors *suspicious*[note 1] of why inadmissible evidence had been introduced ([15]). In two separate studies, these researchers used suspicion to effectively reduce the conviction rates produced by jurors exposed to inadmissible evidence; first, by exposing participants to a newspaper article in which the hypothetical defense attorney questioned the motives of the press in printing incriminating information about the defendant, and second, by embedding an objection to inadmissible testimony in which the hypothetical defense attorney questioned the prosecutor’s motives in introducing the inadmissible evidence. The result was that suspicion made verdicts similar to the conviction rates produced by jurors who had never heard the inadmissible evidence at all. That is, when participants were prompted to question the motives of the proponent of inadmissible evidence, the ironic effects of mental control appear to have been reduced, as was psychological reactance. In legal practice, this intervention could be implemented in much the same way as it was by Fein et al., with a prosecutor or defense attorney questioning the motives of whomever propounded the inadmissible evidence. This could be done through cross-examination or a closing argument. It could not, however, be done by a judge, who must appear impartial to the jury. A judge questioning the motives of either side would provide grounds for appeal.

Another group of researchers reduced conviction rates produced by jurors exposed to inadmissible evidence by giving them a *substantive*, rather than procedural, explanation for why the evidence had been ruled inadmissible ([18]; see also [30]). Specifically, conviction rates were compared between mock jurors who were told that a wiretap was inadmissible because it was illegally obtained (a procedural reason) and mock jurors who were told the wiretap was inadmissible because it was unreliable (a substantive reason). Guilty verdicts were higher in the procedural than the substantive condition. Though this intervention does not fit perfectly in a legal system that frequently requires legal rulings be made for procedural reasons, it could effectively counteract the effects of ironic processes of mental control and psychological reactance in some circumstances. The cognitive processes might be similar to inducing suspicion; when attention is pulled to a reasonable and substantive rationale for ruling evidence as inadmissible, the ironic effects of mental control and psychological reactance may be reduced.

Applied solutions can also be found in the psychological literature regarding compliance. Social psychologists have found many interventions to increase compliance under various conditions. Consistency has long been known to increase compliance with requests that are consistent with a subject’s prior commitment (e.g., [6]). For example, [7] ([7]) studied the tactics car salespeople used to produce compliance from their customers, finding that a technique salespeople called “low-balling” was highly effective in inducing customers to make a purchase decision that they otherwise would be unlikely to make. Specifically, once a person makes an initial decision (e.g., to purchase a specific, bargain-offered car), then even if the circumstances change (e.g., the purchase price goes up for whatever reason or that specific car is unavailable), the person is more likely to stick with their initial commitment (e.g., to purchase a car) than a person would be without having first made a prior commitment. In the context of mock jurors, the consistency literature suggests that asking jurors to *commit* to disregarding any inadmissible evidence at the start of a trial would produce higher rates of compliance with the instruction than simply instructing jurors during the trial, to be consistent with the initial commitment to disregard.

Similar findings emerged in the context of consumer research, where customers were more likely to comply with a purchase request after being asked their opinion about improving the brand ([20]). The putative mechanism driving this effect is that consumers’ subjective perception of their relationship with the organization shifts when their advice is solicited; consumers offering advice feel closer to the organization and are subsequently more likely to transact and engage with it. While the consumer research context may appear dissimilar from jury research, the psychological effect has been endorsed by prominent social psychologists ([5]), and the behavior is likely to persist across different social contexts. In the context of jury research, these findings suggest that asking jurors for *advice* to help future jurors disregard inadmissible evidence at the start of a trial would increase compliance with an instruction to disregard given later during the trial.

## 6. Identifying an Optimized Intervention to Reduce Bias Induced by Inadmissible Evidence

Judges have discretion to decide how to deal with inadmissible evidence. Too often, however, their decisions overlook the compelling data showing that inadmissible evidence has a significant impact on jurors. This psychological impact can disadvantage both plaintiffs and defendants, in both criminal and civil cases. While legal reforms addressing this constitutional issue have been notably slow, a rich foundation of psychological research offers the promise of solutions. In this study, our goal was to identify intervention strategies that can (1) effectively reduce bias induced by inadmissible evidence and (2) be reasonably implemented in U.S. courts. We adopted the multiphase optimization strategy (MOST; [9]), which is a strategic, phased, and data-driven approach to developing an intervention that is maximally effective within the constraints of the setting in which it will be implemented. MOST uses engineering principles (e.g., only include components that have demonstrated a meaningful and favorable effect on the intended outcome) and experimentation (e.g., most commonly, the full factorial experiment) to identify an optimized set of intervention strategies that yield the highest odds of a desired result *and* can be achieved in applied settings.

At the time of this study, MOST has yet to be applied in a legal research setting, making it a novel approach for identifying psycholegal interventions. MOST comprises three phases (preparation, optimization, and evaluation) to develop an optimized intervention (see Figure 1). Within MOST, an optimized intervention is defined as one that balances effectiveness against affordability, scalability, and efficiency. In the *preparation phase*, researchers rely on theory, observation, and empirical evidence to identify intervention strategies that are promising candidates for inclusion in the optimized intervention package. In the *optimization phase,* researchers conduct an optimization randomized controlled trial (optimization RCT) to experimentally evaluate the individual and interactive contributions of each candidate intervention strategy. In the *evaluation phase*, researchers conduct an evaluation RCT to experimentally test the effects of the optimized intervention package against suitable comparisons.

## 7. The Case for Optimization in Legal Interventions

Similarly to engineers building an airplane who use efficient, targeted experiments to ensure the mechanical parts of the machine work as intended prior to taking the first test flight, optimization randomized controlled trials (RCTs) use efficient experimental designs to provide important scientific information about which candidate intervention components are likely to contribute to an overall effect of the intervention package. This approach ensures good use of the resources that are required to conduct a well-powered evaluation RCT. Although it might appear as an additional phase to intervention development research, this methodology can actually conserve time and resources. Without an optimization RCT, inconclusive results from an evaluation RCT offers limited guidance on future actions. A null outcome may indicate either that all intervention components were ineffective, or that some were beneficial while others had adverse impacts. An optimized intervention provides the best expected outcome obtainable within key constraints imposed by the need for efficiency, economy, and/or scalability ([9]). For example, if there is a time or monetary limit on which or how many intervention strategies could feasibly be implemented, researchers identify the combination of intervention strategies that demonstrates the strongest overall effect while also meeting those constraints.

MOST was designed by public health researchers ([9]). In this context, for example, researchers might test which interventions best reduce a person’s odds of contracting COVID-19. Researchers would, based on theory, identify interventions that have the best prospects for achieving the desired outcome. In the COVID-19 example, researchers might consider the effects of masking, social distancing, working from home, and frequent testing. Researchers would then find the combination of interventions that most effectively reduced a person’s odds of contracting COVID-19. While all the examples above likely reduce the odds of COVID-19, some might have to be excluded from public policy for monetary reasons (for example, frequent testing, which became more burdensome after federal funding expired). At the end of this process, COVID-19 researchers would be able to identify the combination of interventions that most effectively reduced a person’s odds of being infected with COVID-19 within budgetary constraints.

MOST may be particularly well suited for legal contexts due to procedural rules that limit the range of permissible interventions. In the current study, we used the entire MOST framework to reduce conviction bias induced by inadmissible evidence. In the following sections, we outlined the activities of each phase.

## 8. The Preparation Phase—Identifying Candidate Intervention Strategies

We first reviewed the prior literature on mock jury studies in laboratory settings to identify intervention strategies that would be good candidates for experimentation. The most important of those studies were detailed earlier in the literature review section of this paper, including [15]’s ([15]) finding that psychological reactance can be reduced by making jurors *suspicious* of why inadmissible evidence had been introduced, and [18]’s ([18]) finding that jurors’ conviction rates could be reduced by giving them a *substantive*, rather than procedural, explanation for why evidence had been ruled inadmissible. We also reviewed the social psychological literature on compliance to identify candidate intervention strategies novel to this context, identifying consistency with prior *commitment* as a key potential intervention, with a theoretically related intervention of asking jurors before they hear evidence to generate *advice* to help other jurors disregard inadmissible evidence (see, e.g., [5]; [6]; [20]).

## 9. Research Overview and Hypotheses

In Study 1, we examined whether any combination of four potential intervention strategies could effectively lower the likelihood of mock jurors convicting based on inadmissible evidence. We used a full factorial randomized design to analyze the individual and combined effects of these strategies to understand their contributions better. While factorial trials are not new—being well documented in the key experimental literature—they are less common in intervention research. However, the MOST framework highlights their value for efficiently determining the causal effects of intervention components. For those less acquainted with this methodology, it is important to note a few essentials: First, statistical power in factorial experiments is influenced by the design’s factors (e.g., presence or absence of an intervention component) rather than the number of observations per group (see Table 1). Every participant’s data contribute to the analysis because the factors are binary. Second, when analyzing data through effect coding, we can interpret interactions as the change in outcome when different factors are combined, without affecting the power to detect these effects compared to the main effects alone. This addresses common concerns about interactions complicating the interpretation of main effects or being too difficult to detect due to insufficient power. Because main effects and interactions in balanced factorial experiments are statistically independent, power for detecting interactions is the same as for main effects in these designs. This efficiency and clarity in interpreting both main and interaction effects highlight the practical advantages of the factorial approach within the MOST framework.

In Study 2, we examined whether adding the optimal intervention package to a judge’s instructions to disregard inadmissible evidence (effectively “standard practice”) would significantly lower conviction rates (i.e., make them similar to the conviction rates by jurors not exposed to inadmissible evidence at all as per the control condition). We used a four-arm randomized controlled trial (RCT) to directly compare the outcomes across different groups. This allowed us to precisely assess the impact of adding the optimized interventions to the conventional approach used by judges, by measuring group differences in conviction rates among the trial arms. Conducting a direct comparison through a four-arm RCT provided a clear, focused analysis of how our intervention could improve upon existing practices by offering a straightforward evaluation of its added value in real-world legal settings.

See Table 1 for a summary review of the differences and implications of the two experimental designs. Both studies were preregistered on the Open Science Framework (https://osf.io/mh6ae/registrations) on 5 July 2023 prior to the collection of any data.

## 10. Study 1

### 10.1. Methods

**Procedure.** We hired Qualtrics to recruit participants through one of their panels to complete an online survey. Qualtrics identifies, screens, and secures commitment from people to become part of their research panels and participate in research, and uses security measures to avoid fraud ([1]). Screening for this survey was based on eligibility for jury service in order to maintain ecological validity, and we also included page timer requirements to ensure participants would stay on each page of the survey long enough to be able to read all the information provided. Qualtrics screened demographics to obtain a sample representative of the US population in order to increase the generalizability of results. Once participants confirmed their jury eligibility and consented to participate, they were randomly assigned to an experimental condition representing 1 of 16 combinations in the factorial experiment. All participants read the trial vignette and jury instructions before being asked to find the defendant guilty or not guilty. This binary outcome variable of verdict (guilty or not guilty) is an ecologically valid reproduction of jury service. Participants were also given the opportunity to provide feedback about the case via an open prompt in order to provide general qualitative data about the experiment. No quantitative analyses were planned for these qualitative feedback data.

**Participants**. Preliminary power analyses indicated that, in order to achieve 80% power to detect an effect size of *d* = 0.33, our minimum target sample size was *n* = 288. A medium effect size was chosen because a small effect size would not be persuasive evidence in making arguments for policy reform and a large effect size would be difficult to detect while excluding small and medium effects that are worthy of consideration. A medium effect size was therefore chosen to balance the interests of persuasive evidence and detectable results. A prior meta-analysis ([35]) reported varying effect sizes from small to large, so the decision was made to identify an effect size of interest.

A demographic analysis of the final sample (*n* = 322) revealed a mean age of 48.97 (range 18–95, *SD* = 18.24). The sample’s gender identity was 23.3% male, 74.5% female, and 2.2% non-binary or gender fluid. Of the 319 participants who responded to the question, 60.9% had never previously served on a jury. The largest proportion of mock jurors (39.5%) had some college education. Politically, the sample was normally distributed, with the highest proportion of respondents (29.5%) self-identifying as neither conservative nor liberal, and the smallest groups being those who identified as very conservative (9.9%) and very liberal (16.1%). The sample closely mirrored census data in terms of race (77.3% White, 11.8% Black, 5.9% Asian/Pacific Islander, 5.0% American Indian/Alaskan Native/other) and ethnicity (82.6% not Hispanic, 17.4% Hispanic).

**Experimental Design.** We used a 2^4^ factorial experiment to test the main and interactive effects of four candidate intervention strategies: (1) inducing suspicion about why the evidence was introduced [*suspicion*], (2) giving a substantive—as opposed to procedural—explanation for why the evidence is inadmissible [*substantive*], (3) asking jurors to commit to disregard inadmissible evidence at the start of the trial [*commitment*], and (4) asking jurors for advice to help future jurors disregard inadmissible evidence [*advice*]. Each candidate intervention strategy had two levels—present or absent. Participants were randomized to sixteen experimental conditions representing all possible combinations of intervention strategies (see Table 2).

**Materials**. The trial vignette described a criminal defendant who had been charged with second-degree murder after shooting the victim in a bar fight (all study materials are available on the Open Science Framework). Though the defendant claimed self-defense, there were no witnesses who could corroborate the claim, and the physical evidence could support either self-defense or intentional second-degree murder. This vignette was designed to introduce multiple points of reasonable doubt without creating an obvious verdict. The trial vignette involved a specific type of inadmissible evidence: prior criminal history. This evidence was chosen because of its prevalence in the American criminal justice system. It is, in fact, so prevalent that there are specific rules of evidence related to the topic (see, e.g., [14]). Additionally, there are substantive reasons for this rule that could be easily adapted to one of the identified interventions (e.g., providing a substantive, as opposed to procedural, explanation for why the evidence is inadmissible).

The four interventions were applied as follows: In the *suspicion* condition, participants read about the defense attorney’s closing arguments, in which the attorney questioned why the prosecution would bring up an old, inadmissible conviction at all, and suggested it was because the state’s case was so weak. In the *substantive* condition, participants received a script in which the judge explained why the evidence was inadmissible (i.e., the defendant’s prior conviction was too old, and the rules of evidence consider old convictions to be less relevant than newer ones). In the *commitment* condition, participants received a script at the beginning of the study, prompting them to commit to disregarding any inadmissible evidence that arose during the course of the “trial.” In the *advice* condition, participants received a script that asked for advice to help future jurors disregard inadmissible evidence.[note 2]

Mock jurors received jury instructions used in the federal courts of the United States. This was done with the use of a Jury Instruction Builder tool authorized by the United States Court of Appeals for the Eleventh Circuit (available at https://pji.ca11.uscourts.gov/ accessed on 5 July 2023). Attorneys use this and similar tools in real cases to determine which jury instructions they will request from the court, so the tool was chosen for its ecological validity.

### 10.2. Study 1 Results: An Optimized Intervention Package

**Preliminary Analyses.** To rule out potential confounds, participants were asked whether they or a close friend/family member had ever been charged with a crime. Neither factor was significantly associated with conviction rates (*p* = 0.67 for self, *p* = 0.67 for a close friend/family member). Contrary to conventional attorney wisdom, respondents’ self-identified politics were also not correlated with conviction rates (*p* = 0.97).

**Verdicts.** We used binary logistic regression analysis with effect coding to estimate the main and interactive effects of the four intervention strategies on binary verdict outcomes (guilty/not guilty). With effect coding, estimates are uncorrelated with balanced cell sizes, and nearly uncorrelated with slight variation in cell sizes (as was the case here). The hypothesis tests are thus independent of one another and no familywise error rate correction (e.g., Bonferroni) was applied ([9]). Among the sixteen possible combinations of the four intervention candidates, the only significant effect was produced by the combination of all four interventions (*b* = 0.25, *p* = 0.03, Exp(B) = 1.28; see Table 3). When a participant received none of the four interventions, there was a 48.24% chance they would vote not guilty, which increased to 60.0% when all four interventions were present (Table 3).

Both theory and statistical analysis indicated that the combination of all four interventions was the optimized intervention package. While the highest-order interaction is sometimes suspected to be spurious, it is possible for each intervention to contribute an insignificant amount of variance that becomes meaningful when all their effects are combined. Our goal was to identify an intervention package composed of all active components that reduce the bias toward conviction associated with inadmissible evidence. Thus, our decision-making focused on identifying the most efficient intervention. There was no monetary cost associated with individual intervention strategies in this study, so budgetary constraints were not considered. The scalable implementation of effective intervention strategies was also not constrained by the number of strategies in the optimized package. Attorneys can make as many pretrial motions as are necessary in order to protect their client’s legal interests (and they actually have a binding legal obligation to do so). Instructing attorneys on the efficacy of interventions could be done through continuing legal education events, which have a fixed cost regardless of the number of interventions included in the curriculum. For these reasons, Study 2 included all four interventions as the optimized intervention package, consistent with an optimization objective of the “all active components” criterion ([9]).

## 11. Study 2

### 11.1. Hypotheses

After identifying the optimized intervention package in Study 1, we predicted that it would be more effective at reducing conviction rates than instructions to disregard on their own, and that both would be more effective at reducing conviction rates than simply admitting inadmissible evidence without objection. In order to directly compare conviction rates, four conditions were created for Study 2: (1) no inadmissible evidence, (2) inadmissible evidence plus standard practice (i.e., judge’s admonition), (3) inadmissible evidence, standard practice, and the optimal intervention package, or (4) inadmissible evidence without any objection.

We planned comparisons for all four groups but focused on three main research questions. First, we assessed the *direct effect of the inadmissible evidence*, comparing the conviction rates of the group that received no inadmissible evidence versus the group that received inadmissible evidence without objection (Groups 1 vs. 4). We expected conviction rates to be higher in the no objection group (Group 4), given that inadmissible evidence is presumed by the law (and demonstrated by empirical research) to increase jurors’ likelihood of conviction.

Second, we assessed the *main effect of standard practice* (i.e., judicial instruction to disregard inadmissible evidence)*,* comparing the conviction rates of the group that received inadmissible evidence plus standard practice of judicial instruction versus the group that received inadmissible evidence without any objection (Groups 2 vs. 4). We did not expect significantly different conviction rates based on prior research showing the low efficacy of the standard practice of judicial instruction. Third, we tested the *overall effect of the optimized intervention package*, comparing the conviction rates of the group that received inadmissible evidence plus standard practice of judicial instruction versus the group that received inadmissible evidence and the optimized intervention package (Groups 2 vs. 3). We expected conviction rates to be higher in the standard practice of judicial instruction group due to the demonstrated efficacy of the optimized intervention package in Study 1.

### 11.2. Methods

**Procedure and Materials.** The procedure for Study 2 was also identical to those in Study 1, except that participants were randomly assigned to one of the four experimental conditions in Study 2. Anyone who participated in Study 1 was ineligible to participate in Study 2 to ensure participants had no prior knowledge of the trial vignette, inadmissible evidence, or proposed interventions. Study 2 used the same materials as Study 1 to reduce the possibility of any confounding variables, whereby changes to the hypothetical criminal charges, evidence, etc., could provide an alternative explanation for differences between the Study 1 results and Study 2 results.

**Participants.** Our preliminary power analysis used the preregistered effect size of *d* = 0.33 and resulted in a minimum sample size of 580 participants. Note that this sample had a much larger cell size (*n* = 145) than in Study 1 (*n* = 18). This is because Study 1 used a factorial experiment, in which each participant affects results on a factor level rather than a group level ([9]). This means that a single participant can affect more than one factor, thereby reducing the overall size of the necessary sample. This efficiency is one of the most practical benefits of MOST, but because Study 2 required a traditional “armed” randomized controlled trial, a full between-group sample size was needed.

A demographic analysis of Study 2’s respondents (*n* = 624) revealed a mean age of 47.29 (range = 18–87, *SD* = 17.58). The sample identified as 30.93% male, 68.1% identified as female, and 0.96% as non-binary or gender fluid. The Study 2 sample was similar to the Study 1 sample in many aspects; most respondents (57.3%) had never served on a jury before. The highest proportion of the sample (37.88%) had some college education and self-identified politically as neither conservative nor liberal (29.49%). Qualtrics again achieved a sample fairly representative of U.S. demographics: 79.3% of respondents identified as White/Caucasian, 13.8% as Black/African American, 3.0% as Asian/Pacific Islander, and 3.8% as American Indian/Alaska Native/other. Ethnically, 84.8% of the sample did not identify as Hispanic and 14.2% did.

**Experimental Design**. We used a four-arm parallel randomized experiment to randomly assign participants to one of four conditions: (1) no inadmissible evidence, (2) standard practice (judge instruction to disregard), (3) optimized intervention package, and (4) no objection. As in Study 1, participants then read the same trial vignette (involving the same inadmissible evidence) before rendering a guilty/not guilty verdict. Participants also had the same opportunity to provide feedback via an open prompt (for qualitative analysis).

### 11.3. Study 2 Results

**Preliminary Analyses.** As in Study 1, verdicts did not differ based on whether the respondent had been charged with a crime (*p* = 0.78) or whether a close friend or family member had been charged with a crime (*p* = 0.43). Self-identified politics were again uncorrelated with verdicts (*p* = 0.67), once more defying attorneys’ folk wisdom about jury selection.

**Verdicts.** Binary logistic regression was used to test the primary hypothesis that a participant’s intervention group would systematically relate to verdict decisions. Group conviction rates were highest to lowest, as follows (Table 4): Group 2 (standard practice of judicial instruction to disregard inadmissible evidence), Group 4 (no objection), Group 1 (no inadmissible evidence), and Group 3 (optimized intervention packages). However, contrary to the hypothesis, the omnibus model showed that there were no significant differences between any of the intervention groups on conviction rates, *X*^2^(3, *n* = 624) = 3.14, *p* = 0.371, Nagelkerke pseudo-R^2^ = 0.007 (Table 5).

We used dummy coding to conduct planned comparisons. Contrary to our hypothesis, the direct effect of the evidence (comparing Groups 1 and 4) was not significant, *b* = 0.051, *SE* = 0.228, *p* = 0.823, Exp(*B*) = 1.05. As predicted, the main effect of standard practice (comparing Groups 2 and 4) was not significant (*p* = 0.76). Jurors who were exposed to inadmissible evidence convicted the defendant at similar rates regardless of whether they were also instructed to disregard. As predicted, the effect of the optimized intervention package was significant compared simply to an instruction to disregard (comparing Groups 2 and 3), *b*(1) = 0.382, *SE* = 0.231, Wald *X*^2^(1) = 2.74, one-tailed *p* = 0.049, Exp(*B*) = 1.47. The finding was only statistically significant using a one-tailed, hypothesized directional test. The size of the effect means that people treated in the standard way—instructed to disregard inadmissible evidence after hearing it—were 1.47 times more likely to convict the defendant than people in the optimized intervention condition. This is a small but meaningful effect.

## 12. Discussion

Although psychologists have consistently and repeatedly found that jurors do not disregard inadmissible evidence when instructed to do so by a judge, the law continues to assume they do. Defendants are left without important constitutional protections at trial and without grounds to challenge improper convictions on appeal. The intervention strategies tested in this study show early promise at helping jurors disregard inadmissible evidence when instructed to do so by a judge. Psychologically, this can reduce bias in juror judgments. Legally, this can prevent wrongful convictions and establish grounds for appealing them.

Standard practice (a judge’s instructions to disregard inadmissible evidence) did not significantly change the odds of a not guilty verdict in either study. Whether jurors cannot or will not disregard inadmissible evidence is a continuing debate, but in legal practice, the distinction is irrelevant. The evidence has demonstrated, yet again, that such instructions do not protect defendants’ due-process protections against inadmissible evidence. While this finding could be due to the failure of the direct effect of the evidence (see “potential limitations” below) it is consistent with decades of prior research in which jurors have repeatedly been found to not disregard inadmissible evidence when they are instructed to do so.

Beyond the replication finding that judges’ admonitions to disregard are ineffective, the findings of these studies suggest preliminary efficacy for what might work better. Consistent with the preregistered hypothesis, we identified a package of interventions based on psychological theory that reduced conviction rates in such jurors across both samples. Though the effect is small, it withstood scrutiny in both Studies 1 and 2. This finding holds promise both for litigators looking to deal with inadmissible evidence in court and for researchers studying potential interventions in the legal system. Theoretically, our findings suggest that a single intervention may not be enough to significantly affect verdicts on its own. Each contributes a small amount of variance that only becomes significant when these small effects are combined. All four interventions may need to be present to meaningfully impact the verdicts of jurors exposed to inadmissible evidence.

### 12.1. Potential Limitations

It is possible that the timing of these interventions could affect their efficacy. The timing of the interventions in this study was determined by logistics; commitment and advice, for example, had to be solicited before the judge instructed jurors to disregard inadmissible evidence (as this is the point at which the intended compliance is sought). The substantive explanation and suspicion, on the other hand, could not be introduced until after the evidence had been ruled inadmissible. It is possible that the timing of these interventions could affect their efficacy. This should be explored in future research, but with the understanding that logistics dictate when the intervention is supposed to take effect and how a trial must proceed.

When comparing conviction rates in light of settled law and extensive prior research, it is surprising that the variance in conviction rates between the groups was low. This low variance could be attributable to potential limitations of the project. First, no mock juror experiment can have the ecological validity of a courtroom. Actual jurors sit in a courtroom for days—and sometimes weeks or months—on end. The average time spent on Study 1 was 14.11 min, with 8.38 min as the average for Study 2. This reduced quantity of deliberation is paired with reduced quality of deliberation as well; when there is no real defendant on trial, there is simply not the same urgency to weigh the evidence appropriately and come to a just verdict. In real trials, juries deliberate as a single body, and this, too, affects the decisions they reach—including cases involving inadmissible evidence (see, e.g., [4]; [21]). In research, these features tend to reduce the effect of interventions tested in the lab. It is worth noting that, in spite of this feature of lab research, the effect of the optimized intervention package (of all four interventions) was significant in both Study 1 (*p* = 0.03) and Study 2 (*p* = 0.049). These findings suggest that these interventions could have an even more powerful effect on real juries that engage in lengthy, meaningful deliberations.

Second, the low variance could be because mock jurors did not understand the basic requirements of conviction. One notable participant reported that they had reasonable doubt, but because the victim was killed, the defendant was guilty. Another juror was even more confused; they voted not guilty, commenting that they would have voted guilty of second-degree murder. (The vignette was, in fact, specific to second-degree murder. This participant seemed to think they were being asked to find the defendant guilty of *first*-degree murder.) For psychological researchers, the concern is that these participants might not have been adequately instructed. For attorneys, the concern is that jurors in real cases can issue similar decisions. The fact is that these responses may well be ecologically valid. Real defendants are convicted by such jurors every day, and they are rarely able to appeal such convictions.

In addition, it is possible the vignette itself could affect the mock jurors’ verdicts. Indeed, this is how the justice system is designed, to allow conviction rates to be affected by the strength of the evidence. While the initial rates of conviction could be instructive, the crucial measure in this study was the change in verdict that occurred with the interventions. Moreover, the baseline verdicts for the vignette (without any interventions) were fairly evenly distributed (51.76% guilty v. 48.24% not guilty in Study 1). This suggests that mock jurors had relatively equal odds of finding the defendant guilty or not guilty. That seems a fair starting point for a jury study focused on how inadmissible evidence might tip the balance, though a more ecologically valid vignette might have conviction odds closer to those faced by actual defendants.

The inadmissible evidence—a prior conviction—was chosen to mirror ([14]) (whose existence demonstrates the legal system’s belief that this evidence will bias jurors toward a guilty verdict). Though this choice was ecologically valid, it did not rise to statistical significance. This limitation illustrates the balance between ecological validity (which improves the external validity of results) and appropriate research methodology (which improves internal validity). This is an ongoing debate for legal researchers, and there is no single right answer that applies universally to all research projects. Nevertheless, future researchers might work to identify inadmissible evidence that would be more persuasive to mock jurors.

Another limitation has to do with the recruitment strategy for participants. We used Qualtrics panels to recruit participants for both studies. While Qualtrics relies on several data quality processes and removes low-quality responses before sending complete data files to scholars who use their research panel services ([1]), there remain some participants in datasets curated by Qualtrics panels that may not pass more intensive data quality screenings ([1]; [24]). Unfortunately, the only other data quality measure we included was time requirements on pages for participants to remain on each page of the study for a certain amount of time before moving forward as a method for preventing skipping through. We did not have further data quality tests in our study measures, such as manipulation checks or attention checks. As such, it is possible that some participants included in our analyses may not have been paying close attention to the research tasks.

### 12.2. Specific Proposals for the Practice and Study of Justice

The findings of this study have important implications for jurists throughout the American legal system. Practicing attorneys should be aware of the persistent effects of inadmissible evidence, and be prepared to brief the court on both the problem and potential solutions that will safeguard their client’s constitutional rights. Trial judges should be open to these suggestions, understanding that decades of scientific evidence have consistently found that jurors do not disregard inadmissible evidence, even when instructed to do so. Trial courts should also consider the persistent effects of inadmissible evidence before dismissing an attorney’s motion for a mistrial on that basis. Finally, appellate courts should consider the effects of inadmissible evidence before broadly dismissing appeals made on those grounds.

Psychological researchers have spent decades considering the problem of instructions to disregard inadmissible evidence. To date, some findings have emerged, but not a sufficient consensus to support systemic changes in the law. Legal systems are inherently resistant to change. In order to overcome this culture, researchers must be united in their calls to action, and their evidence base must be strong. To this end, the proposed solutions must be tested repeatedly. They must demonstrate robustness across a wide range of situations that could apply widely across jurors. The interventions tested in this study have shown some early promise, but they are not yet ready for wholesale legal reform. They must prove their merit by demonstrating replicability across a range of experimental conditions. Other interventions tailored to other theoretical constructs explaining jurors’ inability or unwillingness to disregard inadmissible evidence should also be tested to compare their efficacy. All interventions should be subject to scrutiny through peer review and similar processes. Only by addressing valid scientific criticisms can interventions be ethically endorsed by the scientific community.

## Figures and Tables

**Figure 1 behavsci-15-00007-f001:**
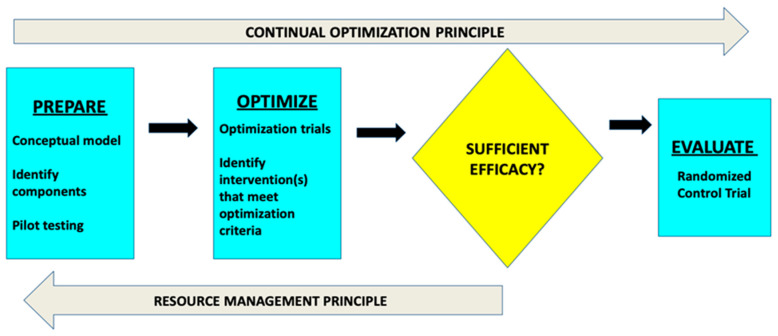
Multiphase optimization strategy.

**Table 1 behavsci-15-00007-t001:** Summary of the study designs and their features in Studies 1 and 2.

Feature	Factorial Design (2 Levels, 4 Factors) Used in Study 1	Parallel Design (4 Arms) Used in Study 2
Description	Evaluates individual and interactive effects of multiple factors on an outcome.	Directly compares the effectiveness of four distinct interventions or treatments.
Sample Size Efficiency	More efficient; fewer participants needed as each contributes to multiple factor analyses.	Requires a larger sample size for adequate power, as each participant is included in only one group comparison.
Power Analysis	*n* = 288 needed to achieve statistical power to detect effect size of *d* = 0.33.	*n* = 580 needed to achieve statistical power to detect effect size of *d* = 0.33.
Practical Implications	Offers insights into how combinations of interventions affect outcomes. Efficient for exploring multiple hypotheses simultaneously.	Provides clear, direct comparisons of intervention effectiveness. Ideal for straightforward evaluation of specific interventions.

**Table 2 behavsci-15-00007-t002:** MOST methodology group assignments for interventions in Study 1.

	1	2	3	4	5	6	7	8	9	10	11	12	13	14	15	16
Standard Practice	Y	Y	Y	Y	Y	Y	Y	Y	Y	Y	Y	Y	Y	Y	Y	Y
Suspicion Induced (Candidate 1)	Y	Y	Y	Y	Y	Y	Y	Y	N	N	N	N	N	N	N	N
Substantive Explanation (Candidate 2)	Y	Y	Y	Y	N	N	N	N	Y	Y	Y	Y	N	N	N	N
Commitment (Candidate 3)	Y	Y	N	N	Y	Y	N	N	Y	Y	N	N	Y	Y	N	N
Advice (Candidate 4)	Y	N	Y	N	Y	N	Y	N	Y	N	Y	N	Y	N	Y	N

Note: Gray “Y” boxes indicate that the group received the intervention. White “N” boxes indicate that the group did not receive that intervention. For instance, Group 1 received a combination of all the interventions, whereas Group 16 received only Standard Practice. Standard Practice = Judicial instruction to disregard inadmissible evidence. Suspicious Induced = Induced suspicion about the reason inadmissible evidence was introduced. Substantive Explanation = Made clear why evidence was inadmissible on substantive grounds. Commitment = Secured prior commitment from mock jurors to disregard inadmissible evidence when instructed. Advice = Participant asked for advice to help future jurors disregard inadmissible evidence.

**Table 3 behavsci-15-00007-t003:** Regression coefficients and significance tests of regression analysis—Study 1.

Intervention(s)	*B*	Significance (*p*)	Rate of Not Guilty Verdict
Intervention Main Effects			
Commitment (C)	0.11	0.354	0.51
Advice (A)	−0.004	0.968	0.48
Substantive Explanation (SE)	−0.15	0.188	0.44
Suspicion (S)	0.15	0.183	0.52
Intervention component two-way interactions			
C*A	0.05	0.651	0.52
C*SE	0.02	0.869	0.48
A*SE	−0.03	0.814	0.44
C*S	−0.01	0.961	0.55
A*S	0.18	0.114	0.57
SE*S	−0.11	0.355	0.46
Intervention component three-way interactions			
C*A*SE	−0.16	0.169	0.44
C*A*S	0.09	0.431	0.62 *(NS)*
C*SE*S	0.07	0.522	0.50
A*SE*S	0.002	0.990	0.49
Intervention component four-way interactions			
C*A*SE*S	0.25	0.030	0.60

**Table 4 behavsci-15-00007-t004:** Group and verdict descriptive statistics—Study 2.

	Guilty	Not Guilty	*n*
Group 1 (No IE)	*n* = 70 (43.5%)	*n* = 91 (56.5%)	161
Group 2 (Standard Practice)	*n* = 72 (46.5%)	*n* = 83 (53.5%)	155
Group 3 (Optimized Package)	*n* = 58 (37.2%)	*n* = 98 (62.8%)	156
Group 4 (No Objection to IE)	*n* = 68 (44.7 %)	*n* = 84 (55.3 %)	152
*TOTAL*	*n* = 268 (42.9%)	*n* = 356 (57.1%)	624

Note: IE = Inadmissible evidence. Standard Practice = Judge’s instruction to disregard inadmissible evidence. Optimized Package = All 4 interventions. *n* = Number of participants.

**Table 5 behavsci-15-00007-t005:** Logistic regression analysis of group comparisons on not guilty verdict—Study 2.

Groups	Regression Coefficient (*b*)	Standard Error	Significance (*p*)	Odds Ratio [*Exp(B)*]
1 (No IE) vs. 4 (No Objection to IE)	0.051	0.228	0.823 *(two-tailed)* 0.412 *(one-tailed)*	1.05
2 (Standard Practice) vs. 4 (No Objection to IE)	−0.069	0.229	0.763 *(two-tailed)* 0.382 *(one-tailed)*	0.93
2 (Standard Practice) vs. 3 (Optimized Package)	0.382	0.231	0.098 *(two-tailed)* 0.049 *(one-tailed)*	1.47

Note: IE = Inadmissible evidence. Standard Practice = Judicial instruction to disregard inadmissible evidence. Optimized Package = All 4 interventions. Groups 1 vs. 4 (row 1) tests the direct effect of the inadmissible evidence. Groups 2 vs. 4 (row 2) tests the main effect of standard practice. Groups 2 vs. 3 (row 3) tests the overall effect of the optimized intervention package.

## Data Availability

The materials and data are available on the Open Science Framework (https://osf.io/mh6ae/, accessed on 1 November 2024).

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
