# Peer review of "Can Jurors Disregard Inadmissible Evidence? Using the Multiphase Optimization Strategy to Test Interventions Derived from Cognitive and Social Psychological Theories"

_behavsci, 2024, doi:10.3390/bs15010007_

Round 1
Reviewer 1 Report
Comments and Suggestions for Authors
This is a beautifully written and presented paper on a fascinating and important topic. The research presented is innovative and well designed and I have no concerns or recommendations for improvement. The only query I had in my mind is whether there was a systematic effect in which the package of interventions affected guilty verdicts - e.g. where some groups more likely to respond to the package - e.g. women, men, younger, older respondents. But this is not really a criticism at all, just curiosity and I am not sure you were able to explore that with the numbers in the experiment. Great paper which should definitely be published.
Reviewer 2 Report
Comments and Suggestions for Authors
The purpose of this manuscript was to examine strategies to successfully get jurors to disregard inadmissible evidence in court. Results revealed that the most optimal intervention consisted of the combined effects of four different strategies—raising suspicion, providing a substantive reason to disregard inadmissible evidence, having jurors commit to disregard inadmissible evidence, and having jurors provide advice to future jurors on how to disregard inadmissible evidence.
Overall, this manuscript offers important contributions to the literature. First, the research examines an important topic—the ineffectiveness of judicial instructions to disregard inadmissible evidence, which is problematic because it can bias jurors’ decision making and ultimately result in wrongful convictions. The present studies aim to identify alternative interventions that are more effective at getting jurors to disregard inadmissible evidence. As such, they provide insight into potential methods to promote fairness and justice in the courtroom. Second, the authors apply a novel, cross-disciplinary approach—the multiphase optimization strategy (MOST)—to study this issue in a new context. The authors also do a nice job making the case for this approach in the context of juror decision-making research. Third, the methodology is sound (e.g., experimental manipulation, high ecological validity).
I do, however, have suggestions for the authors to consider:
Introduction:
- Lines 199-205: The authors describe how some researchers have attempted to reduce psychological reactance by making jurors suspicious of why inadmissible evidence has been introduced. It might be beneficial for the authors to provide an example of this method (e.g., how is suspicion raised? Who might create suspicion—the judge, defense attorney?) to help the reader understand this intervention.
Method:
- Recently, several concerns about data quality have been raised regarding participant recruitment from online panels such as Amazon’s Mechanical Turk (M-Turk) (e.g., see Webb & Tangney, 2022). Although the researchers did not use M-Turk, I am curious if any measures were implemented to ensure the quality of the data (e.g., captchas, methods to remove “bots”)?
- Did the authors use any attention or manipulation checks in the study to ensure participants were paying attention and that their experimental manipulations were effective? If yes, what were the pass rates and was anyone excluded for failing these?
Results:
- Why did the authors employ a one-tailed test to examine verdict differences between the optimized intervention condition and instruction to disregard inadmissible evidence condition (line 564)? This is the only test that was one-tailed, and this raises concerns about their findings considering the effect would no longer be significant if a two-tailed test were used.
Discussion:
- Lines 675-679: The authors note that reactance theory was the primary focus of the interventions in the present studies but that interventions tailored to other theoretical constructs explaining jurors’ inability or unwillingness to disregard inadmissible evidence should also be considered (e.g., Ironic Processes of Mental Control). However, based on the literature reviewed in the introduction, it seemed that the interventions in the present studies were targeting both reactance theory as well as Ironic Processes of Mental Control (e.g., lines 217-219). Can the authors explain how/why their interventions were specifically focused on reactance theory? That is, what elements or constructs did the four interventions target that would mitigate the effects of psychological reactance? Why not also target the effects of mental control?
